# Postoperative Pulmonary Complications after Transcatheter Aortic Valve Implantation under Monitored Anesthesia Care versus General Anesthesia: Retrospective Analysis at a Single Large Volume Center

**DOI:** 10.3390/jcm10225365

**Published:** 2021-11-18

**Authors:** Sang-Wook Lee, Sangho Lee, Kyung-Don Hahm

**Affiliations:** Department of Anesthesiology and Pain Medicine, Asan Medical Center, University of Ulsan College of Medicine, Seoul 05505, Korea; sangwooklee20@gmail.com (S.-W.L.); silzzang15@naver.com (S.L.)

**Keywords:** conscious sedation, transcatheter aortic valve replacement, delirium, postoperative complications, respiration disorders

## Abstract

Few studies to date have assessed the postoperative pulmonary complications after transcatheter aortic valve implantation (TAVI) according to the anesthesia method. The present study aims to compare the effects of general anesthesia (GA) or monitored anesthesia care (MAC) on postoperative outcomes in patients undergoing TAVI. This retrospective cohort study included 578 patients who underwent TAVI through the trans-femoral approach between August 2011 and May 2019 at a single tertiary academic center. The primary outcome was postoperative pulmonary complications, which were defined as the occurrence of one or more pulmonary complications, such as respiratory failure, respiratory infection, and radiologic findings, within 7 days after TAVI. Secondary outcomes included postoperative delirium, all-cause 30-day mortality rate, 30-day readmission rate, reoperation rate, vascular complications, permanent pacemaker/implantable cardioverter-defibrillator insertion, length of stay in the ICU, hospital stay, and incidence of stroke. Of the 589 patients, 171 underwent TAVI under general anesthesia (GA), and 418 under monitored anesthesia care (MAC). The incidence of postoperative pulmonary complications was significantly higher in the GA than in the MAC group (17.0% vs. 5.3%, *p* < 0.001). Anesthetic method significantly affected the occurrence of postoperative pulmonary complications, but not of delirium. ICU stay was significantly shorter in the MAC group, as were operation time, the volume of fluid administered during surgery, heparin dose, transfusion, and inotrope requirements. TAVI under MAC can increase the efficiency of medical resources, reducing the lengths of ICU stay and the occurrence of postoperative pulmonary complications, compared with TAVI under GA.

## 1. Introduction

Transcatheter aortic valve implantation (TAVI) has become a treatment of choice for patients with severe symptomatic aortic valve stenosis at high surgical risk [1,2,3]. In addition, TAVI was recently shown to be a good alternative to surgery in patients at low surgical risk [4,5,6,7], suggesting that the number of patients undergoing TAVI for severe symptomatic aortic valve stenosis will increase. When TAVI was first introduced, general anesthesia (GA) was preferred over monitored anesthesia care (MAC) because GA facilitates more rapid surgical correction of various complications that were due to the lack of experience with the TAVI procedure. Moreover, GA better enabled monitoring by transesophageal echocardiography (TEE) during surgery. Several subsequent meta-analyses and retrospective studies, however, have reported that GA increases patient length of stay in the intensive care unit (ICU) and in the hospital, which may increase medical costs and the inefficiency of medical resources [8,9,10]. Accumulated experience with TAVI has reduced the incidence of complications, with MAC becoming increasingly preferred over GA, especially in reducing the costs reduction related to this procedure. Although few randomized trials have evaluated the impact of anesthetic method during TAVI on postoperative outcome, a recent randomized, multicenter trial showed no differences in postoperative outcome between the GA and MAC groups other than the doses of inotropes and vasopressors administered during the procedure [11,12]. To date, however, the optimal anesthetic method during the TAVI procedure has not been determined, and no studies have evaluated the effect of anesthetic method on the rates of postoperative pulmonary complications (PPCs) after surgery.

PPCs are among the leading causes of increased postoperative mortality and morbidity rates [13,14,15]. However, there is little information on PPCs in patients undergoing TAVI. The present study therefore investigated the effects of the anesthetic method on PPCs in patients who underwent TAVI, as well as evaluating factors affecting these complications.

## 2. Methods

### 2.1. Patients

This study was a large single-center, retrospective study included patients who underwent TAVI through the trans-femoral approach at a single tertiary medical center between August 2011 and May 2019. Patients were excluded if the procedures with an approach other than trans-femoral approach, such as the transapical, transaortic, or trans-subclavian approach, were performed. All clinical data were collected from the electronic medical records system of our institution. This study, which was performed according to the guidelines of the Declaration of Helsinki, was approved by the institutional review board (IRB) of Asan Medical Center (Seoul, Korea, approval number 2020-0838, approval date 27 May 2020, chairperson Professor Moo-Song Lee), which waived the requirement for informed consent due to the retrospective, anonymized nature of this study. 

### 2.2. Perioperative Management According to the Anesthesia Method

Patients underwent trans-femoral TAVI under GA or MAC. Patients were administered GA according to the standard protocol of our institution for GA during TAVI. Beginning before the induction of anesthesia, patients underwent routine monitoring, including by electrocardiography, pulse oximetry, bispectral index (Aspect Medical Systems, Inc., Newton, MA, USA), and cerebral oxygen saturation, with continuous blood pressure monitored through arterial cannulation. Anesthesia was induced by administration of 1.5–2 mg/kg propofol, followed by administration of rocuronium for muscle relaxation after the loss of consciousness was confirmed. Anesthesia was maintained by continual infusion of propofol and remifentanil using a target controlled infusion system. After endotracheal intubation, patients were started on positive pressure ventilation, and a central venous catheter was inserted into the right internal jugular vein. Ventilation during surgery was maintained at an inspiratory to expiratory ratio of 1:2 with a tidal volume of 6–8 mL/kg based on the ideal body weight, with the respiratory rate adjusted to maintain an end-tidal carbon dioxide concentration of 30–35 mmHg. A TEE probe was inserted to evaluate the function of the valve implanted during surgery. 

Prior to the TAVI procedure, valve size and access route were determined by a local multidisciplinary team, consisting of a cardiologist, a cardiovascular surgeon, an echocardiologist, and an anesthesiologist. TAVI was performed according to the standard techniques of our institution by an experienced interventional cardiologist. After valve placement, the function and optimal position of the valve were confirmed by TEE in patients in the GA group, by transthoracic echocardiography (TTE) in patients in the MAC group, and by fluoroscopy in both groups.

After the procedure, patients underwent tracheal extubation in the operating room unless additional mechanical ventilation was required after surgery. The patients were subsequently transferred to the ICU and monitored intensively for various postoperative complications. Patients without major problems in the ICU were transferred to a general ward and discharged.

Before the induction of anesthesia, patients in the MAC group also underwent routine monitoring, including by electrocardiography, pulse oximetry, bispectral index (Aspect Medical Systems, Inc., Newton, MA, USA), and cerebral oxygen saturation, with continuous blood pressure monitored by cannulation of the radial artery. In addition, peripheral venous cannulation was performed for the administration of fluids and drugs such as inotropes during the procedure. The anesthetics for MAC consisted primarily of dexmedetomidine and remifentanil, administered through a target controlled infusion system. Oxygen was administered to patients during surgery through facial masks, and capnographic monitoring was performed to monitor patient respiration during surgery. After the procedure, the patient was transferred to an ICU for intensive monitoring and subsequently moved to a general ward and discharged from the hospital.

### 2.3. Preoperative and Intraoperative Variables

The demographic and clinical characteristics of all included patients were obtained from their electronic medical records. Preoperative factors recorded included patient age, sex, body mass index, American Society of Anesthesiologist (ASA) physical status, smoking history, past medical history (e.g., hypertension, diabetes mellitus, atrial fibrillation, coronary artery disease, chronic kidney disease, cerebrovascular accident, chronic pulmonary disease, chronic liver disease, peripheral vascular disease, malignancy, and dyslipidemia), results of laboratory tests (e.g., hemoglobin and albumin concentrations, and estimated glomerular filtration rate), and preoperative echocardiography findings (e.g., ejection fraction and pulmonary hypertension). Intraoperative variables were also recorded, including the duration of anesthesia, type of anesthetics, total dose of heparin administered, volume of fluids administered (crystalloids and colloids), transfusions, inotropes, and incidence of intraoperative events.

### 2.4. Primary Outcomes and Secondary Outcomes

The primary postoperative outcomes were PPCs. Secondary outcomes included postoperative delirium, all-cause 30-day mortality rate, 30-day readmission rate, reoperation rate, vascular complications, permanent pacemaker (PPM)/implantable cardioverter-defibrillator (ICD) insertion, length of stay in the ICU, hospital stay, and incidence of stroke.

PPCs were defined as the occurrence of one or more pulmonary complications within 7 days after TAVI; these pulmonary complications included respiratory failure, respiratory infection, and radiologic findings associated with atelectasis, pneumothorax, and pleural effusion. Respiratory failure was defined as a partial arterial O_2_ pressure on arterial blood gas analysis below 60 mmHg in room air or O_2_ saturation < 90%, requiring oxygen. Respiratory infection was defined as a post-surgical requirement for antibiotic treatment due to new respiratory symptoms, such as a cough or sputum, and/or fever, and/or findings suspicious of infection on blood tests.

Postoperative delirium was defined as a disturbance in attention and awareness occurring within 5 days after surgery. Because most of the patients in the present study who underwent TAVI were transferred to an ICU, postoperative delirium was evaluated using the Confusion Assessment Method for the ICU (CAM-ICU) and the Richmond agitation-sedation scale (RASS), and by reviewing medical records related to delirium [16,17,18].

### 2.5. Statistical Analysis

Continuous variables were reported as means and standard deviations, and categorical variables as numbers and percentages. Continuous variables were compared in the GA and MAC groups using Student’s *t*-test or the Mann–Whitney *U* test, as appropriate, whereas categorical variables in the two groups were compared using the chi-square test or Fisher’s exact test. Univariate and multivariate logistic regression analyses were performed to identify perioperative risk factors, including anesthetic methods significantly associated with PPCs, the primary outcomes of this study. Variables with *p*-values < 0.05 on univariate logistic regression analysis and those of clinical significance were selected for multivariate logistic regression analysis. All statistical analyses were performed using “R” statistical software (R ver. 3.6.3, R Foundation for Statistical Computing, Vienna, Austria), with a *p*-value < 0.05 considered statistically significant.

## 3. Results

A review of medical records identified 613 consecutive patients who underwent trans-femoral TAVI at Asan Medical Center from August 2011 to May 2019. Of these, 24 patients were excluded; two patients with unknown postoperative outcome due to loss to follow-up after surgery; and 22 patients who underwent TAVI using a trans-apical or trans-aortic or trans-subclavian approach and therefore requiring GA. Finally, 589 patients were included in this study, including 171 who underwent trans-femoral TAVI under GA and 418 who underwent TAVI under MAC (Figure 1). Table 1 shows the demographic and clinical characteristics of these two groups.

### 3.1. Intraoperative Variables

Operation time was significantly longer in the GA than in the MAC group; accordingly, the amounts of fluid and heparin administered during surgery were also higher in the GA (Figure 2, Table 1). Rates of intraoperative blood transfusion and of inotrope administration were also higher in the GA than in the MAC group (Table 1). The incidence of intraoperative events was 6.4% in the GA group and 4.1% in the MAC group, which was higher in the GA group, but there was no statistically significant difference (Table 1).

### 3.2. Primary and Secondary Outcomes

The incidence of postoperative complications was significantly higher in the GA than in the MAC group (21.1% vs. 6.2%, *p* < 0.001, Table 2). Among the sub-grouped variables of PPCs, the GA group showed statistically significantly higher levels than the MAC group in the repository failure, repository infection, and pleural effusion (*p* < 0.02, Table 2). On the other hand, there was no statistically significant difference between the two groups in atelectasis and pneumothorax. Univariable regression analysis of peri-operative risk factors that affected PPCs showed that chronic kidney disease, the status of post percutaneous coronary intervention, cerebrovascular accidents, valve type, anesthetic time, and year of procedure, as well as anesthetic method, were significantly associated with PPCs (Table 3). Multivariable regression analysis that included all the statistically significant variables derived from the univariable analysis, as well as clinically meaningful variables, showed that anesthetic method, chronic kidney disease, the status of post percutaneous coronary intervention, and year of the procedure were the factors significantly associated with PPCs (Table 3).

The incidence of postoperative delirium was lower in the GA (5.3%) than in the MAC (7.2%) group, but the difference was not statistically significant (Table 2). Anesthetic method did not significantly influence the incidence of postoperative delirium.

Among the secondary outcomes, length of stay in the ICU was significantly longer in the GA than in the MAC group (*p*-value = 0.015), whereas there was no statistically significant difference in length of hospital stay between the two groups (Figure 3). In addition, there were no statistically significant differences between the two groups in rates of 30-day readmission, reoperation, 30-day all-cause mortality, vascular complications, PPM/ICD insertion, and postoperative stroke (Table 2).

## 4. Discussion

The present study found that the anesthetic method in patients who underwent trans-femoral TAVI significantly affected the occurrence of PPCs, but not of postoperative delirium. Rates of PPCs were significantly lower in the MAC than in the GA group. In addition, ICU stay was significantly shorter in the MAC group, as were operation time, the volume of fluid administered during surgery, heparin dose, transfusion, and inotrope requirements. However, the rates of 30-day readmission rate, the reoperation, 30-day all-cause mortality, vascular complications, PPM/ICD insertion, and stroke did not differ significantly in these two groups.

Most previous studies have compared aortic valve replacement surgery with TAVI, with fewer studies evaluating the effect of anesthetic technique on PPCs after TAVI [19,20]. The present study found that anesthetic method significantly affected the occurrence of PPCs, even on multivariable regression analysis that included several other co-factors. This finding was not unexpected, as MAC avoids the need for endotracheal intubation and mechanical ventilation [10].

Postoperative delirium, defined as alterations in attention and awareness after surgery [21], is one of the leading causes of negative clinical outcomes, including increased mortality [22]. The incidence of delirium after TAVI has been reported to range from 12% to 53%, with delirium after TAVI having negative clinical outcomes, such as prolonged patient hospitalization and higher mortality rates [22,23,24,25]. The effect of anesthetic method on postoperative delirium in TAVI patients remains unclear [24,26]. The present study found that the anesthetic method did not have a statistically significant effect on the occurrence of postoperative delirium, which was evaluated by reviewing medical records and using CAM-ICU and RASS as assessment tools. The retrospective design of this study may have resulted in an underestimate of the occurrence of delirium, which may have led to an improper evaluation of the effect of anesthetic method on postoperative delirium. Randomized prospective clinical trials are therefore needed to assess the effect of anesthetic method on postoperative delirium in patients undergoing TAVI.

Similar to previous studies, the present study found that anesthetic method affected the lengths of ICU [10,11,27]. This result was consistent with the findings of a recent study based on the National Cardiovascular Data Registry Society of Thoracic Surgeons/American College of Cardiology Transcatheter Valve Therapy Registry, a large-scaled data registry in the United States [28]. However, several other studies have shown conflicting results [11,27]. Because the lengths of stay in the ICU are greatly affected by various medical environments and clinical conditions, additional studies are required to determine the effects of anesthetic methods on these outcomes. 

Because GA generally requires a higher overall dose of anesthetics than MAC, GA likely contributes to hemodynamic instability by negatively affecting cardiovascular function. Although one study reported that inotrope requirement was unrelated to anesthetic method [27], most previous studies have found that patients undergoing TAVI under GA required higher doses of inotropes and/or vasopressors than those in the MAC group due to this hemodynamic instability [10,12,28]. Similar to a recent randomized controlled trial [12], we found that the amount of inotrope administered was higher in the GA than in the MAC group. 

In agreement with most previous studies, the present study found that anesthetic method was unrelated to the 30-day mortality rate. By contrast, a large-scale registry study reported that conscious sedation was associated with a reduced 30-day mortality [11,29]. 

Most previous studies have found that conscious sedation was associated with shorter anesthesia times [11,27,30]. Because MAC does not require time for induction of anesthesia and for extubation, procedure and anesthetic times in patients undergoing TAVI would likely be shorter in the MAC than in the GA group. Although the present results are consistent with this hypothesis, the clinical interpretation of these results is limited because the procedure and anesthetic times were greatly influenced by the physicians’ operative learning curve and specific clinical settings.

Reduced procedure time maximizes the efficiency of use of the operating room, which can reduce total costs for individual patients. The application of MAC for TAVI was found to reduce direct costs by 28% [31]. MAC also enhances the efficiency of use of hospital resources compared with GA, which also affects medical costs.

The present study had several limitations. As a retrospective study, this study has less clinical significance than a prospective randomized trial. Moreover, the distribution of data according to anesthetic method was affected by the timing of the procedure, with MAC performed more often during recent years and GA performed more often during the early stages of the present study. In addition, the number of patients was much higher in the MAC than in the GA group. This prevented the use of methods such as propensity score matching between the two groups. However, to overcome these limitations, the effect of anesthesia method was analyzed by correcting the effect on outcomes over time through multivariable regression analysis. Despite the retrospective design of this study, it still has clinical importance because it is one of the few studies to assess the effects of anesthetic method on the occurrence of PPCs in patients undergoing TAVI. Moreover, this was a large-scaled registry data of non-western patients at a single tertiary medical center. Another limitation of the present study was that factors that could affect the interpretation of results, such as increased operator experience, the development of procedural device, and changes in clinical protocols, could not be excluded.

## 5. Conclusions

In conclusion, the present study showed that the use of MAC rather than GA in patients undergoing TAVI could enhance the efficiency of medical resources by reducing the length of ICU stay and the occurrence of PPCs. These findings suggest that the performance of TAVI under MAC rather than GA can improve clinical outcomes and the efficient utilization of medical resources. Randomized, prospective clinical trials, however, are needed to provide convincing medical evidence for the choice of anesthetic method in patients undergoing TAVI.

## Figures and Tables

**Figure 1 jcm-10-05365-f001:**
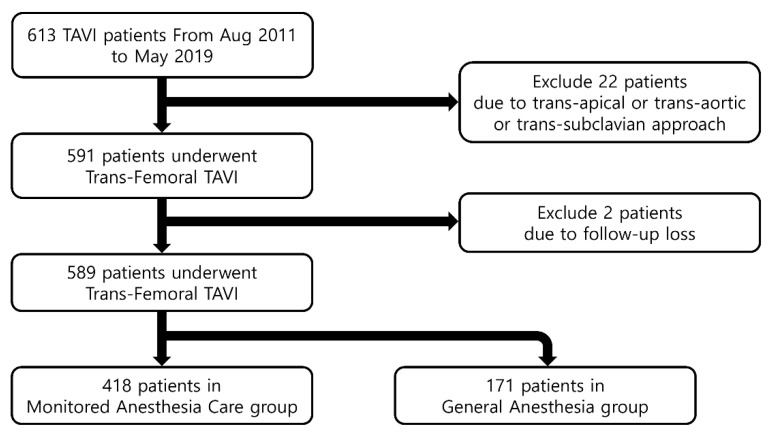
Diagram of the patient flow chart. TAVI, transcatheter aortic valve implantation; Aug, August.

**Figure 2 jcm-10-05365-f002:**
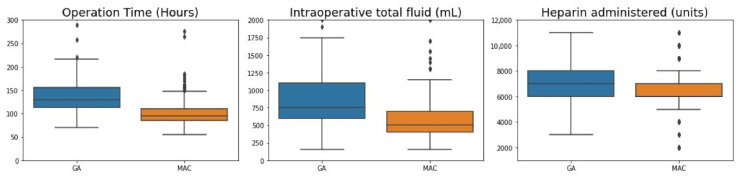
Comparison of intraoperative variables in the GA and MAC groups. GA, general anesthesia; MAC, monitored anesthesia care.

**Figure 3 jcm-10-05365-f003:**
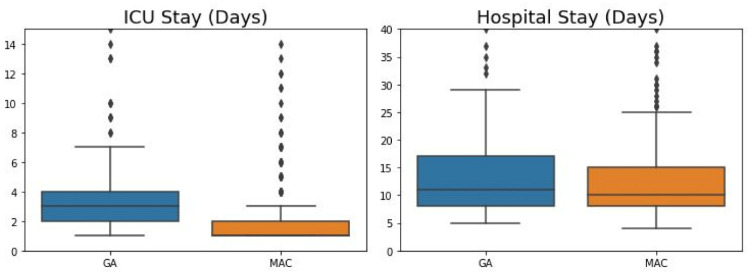
Comparison of hospital stay and length of stay in the intensive care unit in the GA and MAC groups. GA, general anesthesia; MAC, monitored anesthesia care; ICU, intensive care unit.

**Table 1 jcm-10-05365-t001:** Baseline demographic and clinical characteristics of patients who underwent transcatheter aortic valve replacement under MAC or GA. Values are presented as the mean (standard deviation) or number (percentage).

Variables	MAC (*n* = 418)	GA (*n* = 171)	*p*-Value
Age (years)	79.6 (4.9)	78.0 (5.5)	0.001
Males	203 (48.6%)	89 (52.0%)	0.499
Year of procedure, *n* (%)			
2011	3 (0.7%)	13 (7.6%)	<0.001
2012	25 (6.0%)	3 (1.8%)	
2013	34 (8.1%)	12 (7.0%)	
2014	9 (2.2%)	51 (29.8%)	
2015	0 (0%)	44 (25.7%)	
2016	62 (14.8%)	17 (9.9%)	
2017	86 (20.6%)	15 (8.8%)	
2018	143 (34.2%)	12 (7.0%)	
2019	56 (13.4%)	4 (2.3%)	
BMI (kg/m^2^)	24.0 (3.3)	23.6 (3.2)	0.187
Diabetes	124 (29.7%)	61 (35.7%)	0.184
Hypertension	268 (64.1%)	115 (67.3%)	0.529
Atrial fibrillation	59 (14.1%)	21 (12.3%)	0.647
Chronic pulmonary disease	63 (15.1%)	29 (17.0%)	0.654
Chronic kidney disease	47 (11.2%)	26 (15.2%)	0.235
Dialysis	5 (1.2%)	14 (8.2%)	<0.001
eGFR (mg/min/1.73 m^2^)	60.0 (12.8)	60.2 (21.5)	0.908
Hemoglobin (g/dL)	11.6 (1.7)	12.4 (6.9)	0.137
Pulmonary hypertension	99 (23.9%)	57 (33.3%)	0.024
Coronary artery disease	254 (60.8%)	114 (66.7%)	0.212
Previous cardiac surgery	13 (3.1%)	21 (12.3%)	<0.001
Percutaneous coronary intervention	105 (25.1%)	45 (26.3%)	0.843
Preoperative ejection fraction (%)	59.4 (10.1)	56.5 (12.6)	0.007
Preoperative PFT (Normal)	129 (31.4%)	39 (21.9%)	0.025
Preoperative PFT (Obstructive)	61 (14.8%)	22 (12.4%)	0.505
Preoperative PFT (Restrictive)	123 (29.9%)	66 (37.1%)	0.107
Preoperative PFT (Mixed)	78 (19.0%)	30 (16.9%)	0.620
Albumin, g/dL	3.5 (0.4)	3.6 (1.0)	0.157
Dyslipidemia	107 (25.6%)	18 (10.5%)	<0.001
Peripheral vascular disease	12 (2.9%)	2 (1.2%)	0.370
Chronic liver disease	21 (5.0%)	14 (8.2%)	0.200
Malignancy	75 (17.9%)	30 (17.5%)	0.997
Preoperative CVA	114 (27.3%)	18 (10.5%)	<0.001
Emergency	6 (1.4%)	15 (8.8%)	<0.001
ASA class			
II	15 (3.6%)	15 (8.8%)	0.023
III	376 (90.0%)	142 (83%)	
IV	27 (6.5%)	14 (8.2%)	
Valve type			
Balloon-expandable	318 (76.1%)	107 (62.6%)	0.001
Self-expandable	100 (23.9%)	64 (37.4%)	
Intraoperative data			
Propofol (%)	18 (4.3%)	170 (99.4%)	<0.001
Midazolam (%)	373 (89.2%)	11 (6.4%)	<0.001
Dexmedetomidine (%)	405 (96.9%)	2 (1.2%)	<0.001
Transfusion (%)	17 (4.1%)	16 (9.4%)	0.019
Inotropes (%)	48 (11.5%)	36 (21.1%)	0.004
Intraoperative event (%)	17 (4.1%)	11 (6.4%)	0.312

BMI, body mass index; eGFR, estimated glomerular filtration rate; PFT, pulmonary function test; CVA, cerebrovascular accident; ASA, American Society of Anesthesiologists.

**Table 2 jcm-10-05365-t002:** Primary and secondary outcomes. Values are presented as the median (interquartile range) or number (percentage).

Variables	MAC (*n* = 418)	GA (*n* = 171)	*p*-Value
PPCs (%)	26 (6.2%)	36 (21.1%)	<0.001
Respiratory failure (%)	2 (0.5%)	6 (3.5%)	0.009
Respiratory infection (%)	9 (2.2%)	11 (6.4%)	0.019
Atelectasis (%)	9 (2.2%)	9 (5.3%)	0.084
Pneumothorax (%)	0 (0%)	1 (0.6%)	0.290
Pleural effusion (%)	10 (2.4%)	17 (9.9%)	<0.001
Delirium (%)	30 (7.2%)	9 (5.3%)	0.506
30-d mortality (%)	8 (1.9%)	4 (2.3%)	0.752
30-d readmission (%)	22 (5.3%)	17 (9.9%)	0.059
Reoperation (%)	6 (1.4%)	7 (4.1%)	0.062
Vascular complication (%)	29 (6.9%)	14 (8.2%)	0.723
PPM/ICD insertion (%)	39 (9.3%)	13 (7.6%)	0.609
Stroke (%)	11 (2.6%)	9 (5.3%)	0.177

PPCs, postoperative pulmonary complications; PPM, permanent pacemaker; ICD, implantable cardioverter-defibrillator.

**Table 3 jcm-10-05365-t003:** Univariable and multivariable logistic regression analysis of factors associated with postoperative pulmonary complications.

	Univariable		Multivariable	
	OR (95% CI)	*p*-Value	OR (95% CI)	*p*-Value
Anesthetic type (MAC)	0.25 (0.14, 0.43)	<0.001	0.45 (0.25, 0.83)	0.010
Age	1.01 (0.96, 1.06)	0.724		
Sex (female)	0.85 (0.50, 1.44)	0.544		
BMI	0.99 (0.92, 1.07)	0.574		
DM	1.44 (0.82, 2.46)	0.192		
HTN	1.62 (0.91, 3.02)	0.1124		
A-fib	1.09 (0.48, 2.21)	0.821		
CPD	1.04 (0.48, 2.06)	0.907		
CKD	2.06 (1.02, 3.93)	0.033	3.29 (1.50, 7.01)	0.002
Dialysis	2.35 (0.65, 6.74)	0.140		
CAD	1.10 (0.64, 1.94)	0.726		
s/p Cardiac op	2.36 (0.91, 5.40)	0.055		
s/p PCI	0.46 (0.21, 0.92)	0.040	0.36 (0.15, 0.76)	0.012
Preoperative CVA	0.41 (0.17, 0.86)	0.031	0.84 (0.32, 1.96)	0.695
Valve type (SE)	2.37 (1.38, 4.05)	0.002	1.46 (0.80, 2.62)	0.213
Anesthetic time	1.01 (1.00, 1.01)	<0.001	1.00 (1.00, 1.01)	0.149
Transfusion	1.98 (0.72, 4.72)	0.147		
Year	0.70 (0.62, 0.79)	<0.001	0.74 (0.64, 0.85)	<0.001

MAC, monitored anesthesia care; BMI, body mass index; DM, diabetes mellitus; HTN, hypertension; A-fib, atrial fibrillation; CPD, chronic pulmonary disease; CKD, chronic kidney disease; CAD, coronary artery disease; s/p, status post; PCI, percutaneous coronary intervention; CVA, cerebrovascular accident; SE, self-expandable.

## Data Availability

The data of this study are available from the corresponding author upon reasonable request.

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
