# Peer review of "Postoperative Pulmonary Complications after Transcatheter Aortic Valve Implantation under Monitored Anesthesia Care versus General Anesthesia: Retrospective Analysis at a Single Large Volume Center"

_jcm, 2021, doi:10.3390/jcm10225365_

Round 1

Reviewer 1 Report

In this retrospective study, the authors compared the incidence of complications related to transcatheter aortic valve implantation (TAVI) between patients who underwent general anesthesia (GA) and those who had monitored anesthesia care (MAC). They found that postoperative pulmonary complications (PPCs) occurred more in the GA group than in the MAC group, although the rates of 30-day all-cause mortality, vascular complications, permanent pacemaker implantation/implantable cardioverter-defibrillator insertion, and stroke did not differ significantly between the two groups. This article is unique in reporting the incidence of PPCs along with other commonly reported complications in patients undergoing TAVI, which is clinically relevant. However, there are several issues that authors might wish to address as follows:

  1. In Figure 1, it seems that the authors compared the clinical events between MAC group and GA group in patients who had successfully received TAVI under either of the aesthetic methods after excluding patients who had had events during the procedures. However, a switch from MAC to GA owing to intra-procedural events or conversion from trans-femoral to trans-apical or trans-aortic are also important clinical events, which cannot be captured by this study method. For example, even if intra-procedural events had occurred more in the MAC group than the GA group, this downside of the MAC group would have been overlooked by the method the authors took in this study. I would suggest not excluding patients who had a change in the anaesthetic method and conversion from trans-femoral to trans-apical or trans-aortic approach. At least, the events rate during procedure both in the MAC and GA groups need to be reported.
  2. Is the decrease of PPCs attributable solely to the anaesthetic method (MAC)? In Table 3, year is also significantly related with the reduction of PPCs, which indicates that the TAVI procedure has matured over time and several factors part from the anaesthetic method might have contributed to the reduction of complications, for instance, newer devices, gain of experience among staff, and so on. The introduction of MAC to TAVI must have contributed to the improvement of clinical outcome, but it might be just one of the contributing factors.
  3. In Figure 1, 22 patients appear to have undergone trans-apical or trans-aortic approach. Did no one have other approach such as trans-subclavian approach?
  4. It would be better to provide information about cardiac and pulmonary function, for instance echocardiographic findings and pulmonary function test that will influence the clinical outcomes.
  5. The details of PPCs need to be clarified. PPCs in this study consists of several pulmonary complications, each of which has different clinical significance.
  6. In table 3, not all variables of statistical significance in the univariable analysis are included into the multi-variable analysis. Why?
  7. It would be helpful for readers to appreciate the transition from GA to MAC over the period of time, if the numbers and proportions of GA and MAC in each year during the study period.
  8. The incidence of PPCs occurring in 21.1 % of the GA group appears to be a bit too high. Is this the case to other types of surgeries that required GA in the same period or is there any specific reason for this high incidence in the patients who underwent TAVI?
  9. In table 2, are there any reasons for the less incidence of 30-day readmission and re-operation in the MAC group than the GA group? This might be again due to the maturity of the procedure as mentioned in the second comment.

Author Response

Response to Reviewer 1 Comments

In this retrospective study, the authors compared the incidence of complications related to transcatheter aortic valve implantation (TAVI) between patients who underwent general anesthesia (GA) and those who had monitored anesthesia care (MAC). They found that postoperative pulmonary complications (PPCs) occurred more in the GA group than in the MAC group, although the rates of 30-day all-cause mortality, vascular complications, permanent pacemaker implantation/implantable cardioverter-defibrillator insertion, and stroke did not differ significantly between the two groups. This article is unique in reporting the incidence of PPCs along with other commonly reported complications in patients undergoing TAVI, which is clinically relevant. However, there are several issues that authors might wish to address as follows:

1. In Figure 1, it seems that the authors compared the clinical events between MAC group and GA group in patients who had successfully received TAVI under either of the aesthetic methods after excluding patients who had had events during the procedures. However, a switch from MAC to GA owing to intra-procedural events or conversion from trans-femoral to trans-apical or trans-aortic are also important clinical events, which cannot be captured by this study method. For example, even if intra-procedural events had occurred more in the MAC group than the GA group, this downside of the MAC group would have been overlooked by the method the authors took in this study. I would suggest not excluding patients who had a change in the anaesthetic method and conversion from trans-femoral to trans-apical or trans-aortic approach. At least, the events rate during procedure both in the MAC and GA groups need to be reported.

  • First of all, thank you for your valuable review of our research. Originally, the reason for excluding patients who had a change in anesthesia method during the procedure was because it was judged that it was difficult to fully evaluate the effect of anesthesia method on pulmonary complications, the main outcome of our study, due to the change in anesthesia method. However, I fully understand the point that it may result in underestimating the occurrence of events in the MAC group because there may be changes in the anesthesia method due to events occurring during surgery. Therefore, 11 patients who had a change in anesthesia method due to the occurrence of an event during surgery were not excluded and included in the analysis of the results. In this way, the manuscript for the change in exclusion criteria was revised. You can see that in the line23 of page 2 and the line 23 of page 4. First of all, I am very sorry for the error of figure1 which is different from the intention. The figure 1 was modified because there was a part that could be misunderstood as a change in approach as the reviewer thought in figure 1. You can see that in Figure 1 of page 5. The patients who took a transapical or transaortic approach did not convert the procedure approach due to the occurrence of an event during surgery, but decided the approach in advance according to the blood vessel condition before surgery. Also, looking at previous studies, the patients who were treated with only one approach (trans-femoral approach) were analyzed postoperative outcomes according to the change of anesthesia method (Hendrik Stragier, et al. JCVA 33(2019), 3283-3291). So, I think it is reasonable to analyze only the trans-femoral approach that has been implemented the most frequently. However, if you has a different opinion on this point, we will proceed with the analysis including a different approach (trans-aortic or trans-apical approach). Additionally, the intraoperative event rate in the MAC and GA groups was added to table 1. You can see that in Table 1 of page 6. These parts were described in addition to the manuscript (line 7 of page 7).

2. Is the decrease of PPCs attributable solely to the anaesthetic method (MAC)? In Table 3, year is also significantly related with the reduction of PPCs, which indicates that the TAVI procedure has matured over time and several factors part from the anaesthetic method might have contributed to the reduction of complications, for instance, newer devices, gain of experience among staff, and so on. The introduction of MAC to TAVI must have contributed to the improvement of clinical outcome, but it might be just one of the contributing factors.

  • First of all, thank you for your good opinions. As you said, I fully agree with the opinion that there may be improvements in postoperative outcomes over time due to advances in proficiency and technology in the procedure. However, through the results of our study, it was confirmed that even if the postoperative outcome improvement over time was excluded, the difference in anesthesia methods had an effect on the occurrence of PPCs. From the results of Table 3, it was found that MAC had a statistically significant effect on the reduction of PPCs in multivariable regression analysis as well as univariable regression analysis. In multivarible regression analysis, it can be said that the choice of MAC affected the decrease of PPCs because it was statistically significant even when the year of procedure and anesthesia method were put into the regression model together. This is the same as correcting the effect on the decrease of PPC according to the year of procedure. Therefore, the results of our study can be said that the the anaesthetic method (MAC) had a statistically significant effect on the decrease of PPCs. An explanation for this part was added to the discussion part of the manuscript. You can see that in the line 31 of page 10.

3. In Figure 1, 22 patients appear to have undergone trans-apical or trans-aortic approach. Did no one have other approach such as trans-subclavian approach?

  • First of all, thank you for the good point. As a result of a detailed review of the data again, 13 out of 22 patients underwent transapical approach, 8 patients underwent trans-aortic approach, and 1 patient underwent trans-subclavian approach. So, we modified some of the confusing parts of the text. The revised parts are as follows. ( the line 25 of page 2, the line 28 of page 4, Figure 1 in page 5 )

4. It would be better to provide information about cardiac and pulmonary function, for instance echocardiographic findings and pulmonary function test that will influence the clinical outcomes.

  • Thank you for the good suggestion. First, the preoperative ejection fraction, a representative indicator of cardiac function, was included in Table 1. In addition, the pulmonary function test result report, which can indicate a pulmonary function, was also included in table 1 with statistical values for each group. You can see that in Table 1 of page 6.

5. The details of PPCs need to be clarified. PPCs in this study consists of several pulmonary complications, each of which has different clinical significance.

  • Thank you for your good suggestion. Table 2 was modified by dividing the differences seen in each group according to the sub-grouped items of the PPC. You can see that in Table 2 of page 7. Related content was added to the manuscript. You can see that in the line 12 of page 7.

6. In table 3, not all variables of statistical significance in the univariable analysis are included into the multi-variable analysis. Why?

  • First of all, the study dataset changed due to the change in the study exclusion criteria according to your first suggestion (No.1 review item of reviewer 1). Table 3 was revised to reflect the new statistical results, and in the new Table 3, only significant variables in univariable analysis were used for multivariable analysis by reflecting the opinions you gave. You can find revised Table 3 in page 8.

7. It would be helpful for readers to appreciate the transition from GA to MAC over the period of time, if the numbers and proportions of GA and MAC in each year during the study period.

  • As you suggested, the numbers and proportions of anesthesia methods for each procedure year were added to table 1. You can see that in Table 1 of page 6. I would appreciate it if you could let me know whenever you have any additional points to point out in the revised part. Thank you for the good suggestion.

8. The incidence of PPCs occurring in 21.1 % of the GA group appears to be a bit too high. Is this the case to other types of surgeries that required GA in the same period or is there any specific reason for this high incidence in the patients who underwent TAVI?

  • Most of the patients receiving TAVI in our hospital are elderly, with an average age exceeding 79. In addition, most of the patients have decreased pulmonary function, with the proportion of normal people less than 30% in the pulmonary function test performed before surgery. Additionally, TAVI is often selected in patients with a high risk of surgical methods (SAVR). Therefore, due to these factors, the PPC incidence rate may be higher than that of other general anesthesia groups.

9. In table 2, are there any reasons for the less incidence of 30-day readmission and re-operation in the MAC group than the GA group? This might be again due to the maturity of the procedure as mentioned in the second comment.

  • As a result of re-analysis with a new dataset, there was no statistically significant difference between the two groups in the 30-day re-hospitalization rate and re-operation rate ( p-value of 30 d readmission : 0.059, p-value of re-operation : 0.062 ). However, in the GA group, the two indicators were higher than MAC. This may be related to the maturity of operator's experience as you mentioned. However, through this study, it seems that the influence of the anesthetic method cannot be clearly explained.

Reviewer 2 Report

Lee et al. investigated the incidence and risk factors of postoperative pulmonary complications in 578 patients undergoing TAVI at a single center. Overall, it is an interesting retrospective cohort study with strong clinical significance and relevance. My only suggestion is the authors should present the key data more clearly.

Author Response

  • Thank you for your good opinion. We accepted the opinions of several reviewers and modified the data to make it more clearly to readers. If there is anything lacking in our paper, please feel free to give us additional suggestions and we will reflect them in the text. In addition, it is possible to provide the entire research data upon appropriate request through the corresponding author. This fact was described as a data availability at the end of the manuscript. Once again, thank you for your valuable review.

Reviewer 3 Report

Thank you for the opportunity to review this interesting paper. This paper is comparing complications after TAVR according the the anesthesia method, general anesthesia (GA) vs monitored anesthesia care (MAC). The author is especially looking at the incidence and determinants of post-operative pulmonary complications and delirium.

Overall the manuscript is well written but as a reader I wish to make some suggestions that I believe will improve the manuscript further.

Abstract:

The authors should clarify this study is comparing complications depending on the anesthesia method. And they need to explain the definition of post-operative pulmonary complications and describe those incidence and determinants in the abstract as well, not only in the text.

Overall:

  1. This is a comparative study of complications after TAVR depending on the anesthesia method, GA vs MAC, not primarily looking at the post-operative pulmonary complications. Please consider change the title.

  1. Please clarify how you selected the anesthesia method. Did the anesthesia method differ depending on the year of the procedure? Did the operator experience affect the selection of the anesthesia method? If the GA was hired in 2019, were there any specific reasons, such as high risk? How about the surgical risk of TAVR patients? The risk of SAVR affected the anesthesia method?
  2. Can you disclose the reasons for the re-operation and why 30-day readmission or re-intervention were affected by the anesthesia method?

Author Response

Thank you for the opportunity to review this interesting paper. This paper is comparing complications after TAVR according the the anesthesia method, general anesthesia (GA) vs monitored anesthesia care (MAC). The author is especially looking at the incidence and determinants of post-operative pulmonary complications and delirium.

Overall the manuscript is well written but as a reader I wish to make some suggestions that I believe will improve the manuscript further.

Abstract:

The authors should clarify this study is comparing complications depending on the anesthesia method. And they need to explain the definition of post-operative pulmonary complications and describe those incidence and determinants in the abstract as well, not only in the text.

  • First of all, thank you for your valuable review. As you mentioned, the fact that complications according to the anesthesia method are compared, the definition of PPC, and incidence were added to the Abstract section to correct and supplement them. If there is anything additional to correct, please feel free to mention it and we will reflect it.

Overall:

1. This is a comparative study of complications after TAVR depending on the anesthesia method, GA vs MAC, not primarily looking at the post-operative pulmonary complications. Please consider change the title.

  • According to the reviewer's point, the title of the paper was changed to "Postoperative pulmonary complications after transcatheter aortic valve implantation under monitored anesthesia care versus general anesthesia: retrospective analysis at a single larger volume center". If there are any additional corrections to the changed title, please let us know and we will reflect it. Thank you for the good suggestion.

2. Please clarify how you selected the anesthesia method. Did the anesthesia method differ depending on the year of the procedure? Did the operator experience affect the selection of the anesthesia method? If the GA was hired in 2019, were there any specific reasons, such as high risk? How about the surgical risk of TAVR patients? The risk of SAVR affected the anesthesia method?

  • Despite the limitations of our study being a retrospective analysis of the TAVI registry of our institution, we were able to confirm various interesting results through data. According to an analysis of general anesthesia cases conducted in 2019, patients who underwent general anesthesia were not due to the patient's comorbid disease or surgical risk, but to difficulties in performing MAC due to severe cognitive impairment or needs of trans-esophageal echocardiography (TEE) during surgery. The patient's comorbidities or risk of surgery may affect the choice of anesthesia methods, but that is not a major and an only variable for selection of anesthetic method. The anesthesia method is selected in consideration of various factors such as anesthesiologist's preference, proficiency in the procedure, use of TEE, difficulty in MAC anesthesia, and comorbidities of patients. Although the distribution of anesthesia methods is uneven depending on the time of the procedure, as a result of analyzing a relatively large number of data in a single institution, it was confirmed that the anesthesia method affects PPCs, even though the effect of the time is corrected through multivariate regression analysis. An explanation for this part was added to the discussion part of the manuscript. You can see that in the line 31 of page 10.

3. Can you disclose the reasons for the re-operation and why 30-day readmission or re-intervention were affected by the anesthesia method?

  • As a result of re-analysis with a new dataset, there was no statistically significant difference between the two groups in the 30-day re-hospitalization rate and re-surgery rate ( p-value of 30 d readmission : 0.059, p-value of re-operation : 0.062 ). However, in the GA group, the two indicators were higher than MAC. However, since it is not a statistically significant result, it is difficult to explain the appropriate association. It seems that the influence of the anesthetic method cannot be clearly explained. The contents of the manuscript according to the new statistical results have been revised. You can see that in Table 2 of page 7, the line 9 of page 8 and the line 9 of page 9.

Round 2

Reviewer 1 Report

All the issues I raised were solved properly. I would like to congratulate the authors for all their efforts to resolve these issues in a very short period of time.

Reviewer 3 Report

Thank you for the opportunity to review this interesting study. The author sufficiently responded all the issues I have pointed out.